# Improvement and Validation of NASA/MODIS NRT Global Flood Mapping

**Li Lin, Liping Di \*, Junmei Tang, Eugene Yu, Chen Zhang, Md. Shahinoor Rahman** **, Ranjay Shrestha and Lingjun Kang**

Center for Spatial Information Science and Systems, George Mason University, Fairfax, VA 22032, USA;
llin2@gmu.edu (L.L.); jtang8@gmu.edu (J.T.); gyu@gmu.edu (E.Y.); czhang11@gmu.edu (C.Z.);
mrahma25@gmu.edu (M.S.R.); ranjay.shrestha@gmail.com (R.S.); lkang3@gmu.edu (L.K.)
\* Correspondence: ldi@gmu.edu; Tel.: +01-703-993-6114

**Abstract:** The remote-sensing based Flood Crop Loss Assessment Service System (RF-CLASS) is a web service based system developed and managed by the Center for Spatial Information Science and Systems (CSISS). The system uses Moderate Resolution Imaging Spectroradiometer (MODIS)-based flood data, which was implemented by the Dartmouth Flood Observatory (DFO), to provide an estimation of crop loss from floods. However, due to the spectral similarity between water and shadow, a noticeable amount of false classification of shadow can be found in the DFO flood products. Traditional methods can be utilized to remove cloud shadow and part of mountain shadow. This paper aims to develop an algorithm to filter out noise from permanent mountain shadow in the flood layer. The result indicates that mountain shadow was significantly removed by using the proposed approach. In addition, the gold standard test indicated a small number of actual water surfaces were misidentified by the proposed algorithm. Furthermore, experiments also suggest that increasing the spatial resolution of the slope helped reduce more noise in mountains. The proposed algorithm achieved acceptable overall accuracy (>80%) in all different filters and higher overall accuracies were observed when using lower slope filters. This research is one of the very first discussions on identifying false flood classification from terrain shadow by using the highly efficient method.

**Keywords:** MODIS; false water classification; terrain shadow; noise reduction

---

## 1. Introduction

Flood, the most widespread and frequent natural disaster in the world, has profound impacts on populations worldwide with significant loss of life and property each year [1,2]. The economic damage caused by significant floods reached at least \$166 billion worldwide between 1995–2015, which accounts for more than 20% of the total natural disaster damages [3]. An accurate estimation of flood extent not only meets the growing need to assess the damaged areas during the emergency events for flood detection and management, but is also essential to establish the flood early warning system for mitigating its catastrophic effects [4,5].

Remote-sensing data, in conjunction with other geospatial information, has been recognized as an effective means to provide a spatially distributed flood area in the timely monitoring of flood events [6–12]. Early applications focused on the flooded area delineation using optical remote sensing such as SPOT XS [13], the Landsat Thermatic Mapper (TM)/Multispectral Scanner (MSS) [14,15], and the National Oceanic and Atmospheric Administration Advanced Very High Resolution Radiometer (NOAA-AVHRR) [16]. Much of these pioneering works developed detection methods based on the spectral characteristics of studying objectives during floods. This initial detection might not be universally effective because the natural condition of flooding varies locally and weather

conditions and cloud cover also minimize the effects of optical remote sensing. A microwave sensor, such as the synthetic aperture radar (SAR) light detecting and ranging (LIDAR), has been used as an excellent tool to monitor flood in bad weather condition due to its capability in penetrating clouds as an active sensor [17,18]. However, this data still faces obstacles in accurately delineating the flood extension, particular for the inundated with the wind-induced ripples or tree-mixed surface problem [18]. Moreover, the incident angle and consequent variation in back scatter as well as its discontinuous acquisition pose difficulties in the quick response from flooding using active sensors [17].

The MODerate Resolution Imaging Spectroradiometer (MODIS) onboard the National Aeronautics and Space Administration's (NASA) Terra and Aqua satellites, provides unprecedented information globally with high temporal and spectral resolution which can be used for regional and global scale flood monitoring in near real-time [4,19,20]. Although the spatial resolution of MODIS is relatively coarse (~250m) compared with other satellite sensors, MODIS is still one of the most significant sources to provide support to flood assessment [10]. Many algorithms have been developed to identify the flooded area, including the index difference algorithm between the land surface water index (LSWI) and vegetation indexes (VI) [21], the vegetation cover conversion algorithm to detect the flood season between July and September [22], and the decision tree approach to derive the water fraction and flood maps [18].

Early flood warning and real-time monitoring systems play a significant role in flood risk reduction and disaster response decision. [23] developed an initial satellite-based global flood monitoring system (GFMS) which is operationally available at the Tropical Rainfall Measuring Mission (TRMM) website (http://trmm.gsfc.nasa.gov/). Based on MODIS satellite data, Dartmouth Flood Observatory (DFO) developed a near real-time (NRT) global flood mapping 3-day product to monitor flood disaster all over the world [24]. Although these satellite-based global flood monitoring and forecasting systems are currently operating, their reliability for decision-making applications needs to be addressed and validated [25]. In particular, the shadow, one of the most common types of error encountered in remotely sensed data, has generated confusion and misclassification in detecting flooded area [26–28].

The aim of the present study is to improve and validate the MODIS-derived global flood mapping products. Specifically, we will create a methodology that can be used to detect and remove the shadow-caused misclassification with the focus on the topographic shadow. This methodology was designed to improve the existing NRT global mapping product. Using NASA/DFO as an example, various flood products were used to test the algorithm at different regions in the United States (Iowa County, Iowa; Fremont County, Colorado; and Southeastern Idaho). The accuracy of the proposed algorithm was assessed using inundation data from Valley County, Idaho.

The NASA/DFO NRT global flood product is generated based on the ratio of MODIS 250-m band 1 (red) and band 2 (near infrared) as well as a threshold on band 7 (shortwave infrared) and composited over 2-day (2D2OT), 3-day (3D3OT), and 14-day(14×3D3OT) periods [29]. This product has been applied widely in flood forecasts [25], flood mapping [30], and flood crop loss assessment [31]. The DFO operational flood product has the following two advantages compared with other flood products: daily temporal resolution and global coverage. However, due to the high revisit circle, the spatial resolution is relatively coarse. In addition, poor cloud penetration and the spectral similarity between water and shadow also introduced lot of misclassifications in the product. Previous evaluation of the flood products indicated that extreme terrain, terrain shadow and cloud shadows are the primary source of errors in flood detection [25]. Although DFO has adopted various algorithms such as using multi-temporal images, solar azimuth and zenith angle to remove false classifications, developing an algorithm to minimize the effect of the shadow is still necessary for future upgrades and enhancements in this product.

Cloud shadow is one of the significant concerns in moderate-resolution optical remotely sensed data and the exclusion of cloud shadow pixel is an essential pre-processing step for various applications such as flood detection, land-use change detection, resource management [28]. To detect and

remove the cloud shadow, two types of method have been developed in the past. Generally, cloud shadow is detected either based on its spectral characteristics (e.g., surface reflectance or brightness temperatures) [32,33] or geometry correlation [34,35]. For the first method, many algorithms have been developed, including the image-difference method [36], the iterative self-organizing data analysis (ISODATA) technique [37], the multi-channel threshold [38], and the closest spectral fit methodology [39].

Compared to the spectral-based method, the geometry-based method investigates the geometric relationship among clouds, sun, and satellite to predict the positions and area of cloud shadows given the solar azimuth and zenith angle, viewing geometry, cloud altitude and cloud-edge distribution [32,40]. This method has been applied to most of the moderate resolution satellite images from the NOAA-AVHRR product [32,41], MODIS [40], and the visible/infrared imager/radiometer suite (VIIRS) [35]. Generally, the geometry-based method requests cloud phase, cloud type, cloud top/bottom height to obtain accurate cloud shadow which might create difficulties for rapid-response application such as flood detecting and monitoring [40]. Since cloud moves continually and the cloud shadow rarely stays in the same locations, some flood mapping products, e.g., the NASA/DFO MODIS NRT global flood mapping product, use time-series images to remove cloud shadow noise effectively.

Terrain shadow is another challenge for land products derived from moderate-resolution optical satellite data [42,43], particular for high-quality flood detection due to the substantial similarity in spectral characteristics [28]. By contrast with cloud shadow, which can be removed by defining the continuous exposure, the terrain shadow is hard to remove automatically, and many deep terrain shadows were misclassified as floodwater [44].

Most of the de-shadow methods still are based on the spectral characteristics [43,45] or band ratio [46] and focused on high-resolution satellite imagery. However, few studies provide practical solutions to the terrain shadow problem in the global near-real-time products derived from coarse-to-moderate resolution satellite imagery. For example, the terrain shadows have similar spectral with floodwater while the former always present in the mountainous areas while the latter still accumulate in relatively flat areas. In the NASA/DFO product, almost 26% of the total flooded area was delineated in the area with more than 5° and more than 15% are mapped in the area with more than 10° (Figure 1). Considering the high frequency of this misclassification, it is necessary to remove the terrain shadow to improve the global near-real-time flood detection maps.

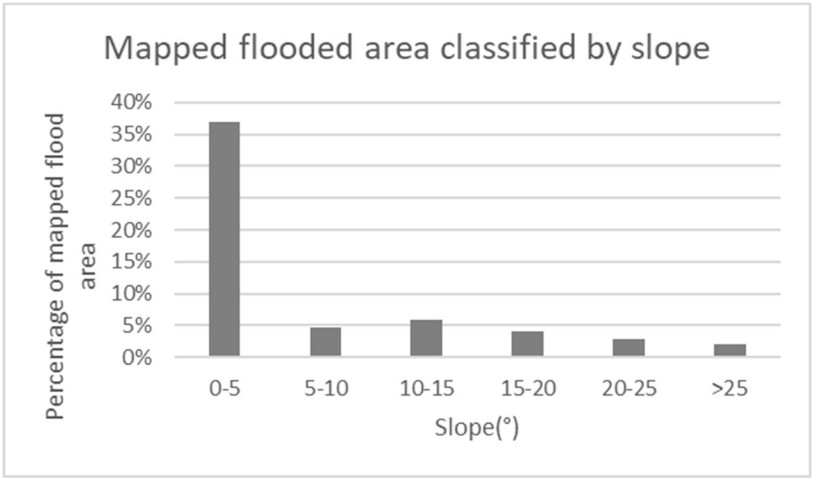

**Figure 1.** Mapped flooded area in the United States from 2011–2015 by National Aeronautics and Space Administration/Dartmouth Flood Observatory (NASA/DFO) near real-time (NRT) global flood product.

## 2. Materials and Methods

### 2.1. Study Area and Data Processing

In this study, two areas with different topographic characteristics in different states were selected for experimental purposes: Iowa County in Iowa to represent the flat land and Fremont County in Colorado to represent mountain region (Figure 2). An experiment on Iowa County was designed to see the performance of the proposed algorithm when flood occurs in flat areas, and Fremont County was chosen to see the filter's result for floods in mountains. To further test the algorithm, we selected the southeastern Idaho region with both flat and mountainous region to see the performance of algorithm in large scale. In additional to represent the differences in geomorphology, data from various time periods were collected: Iowa County and Fremont County experienced significant floods for the selected study timeframe, while southeastern Idaho has no flood event during the study period.

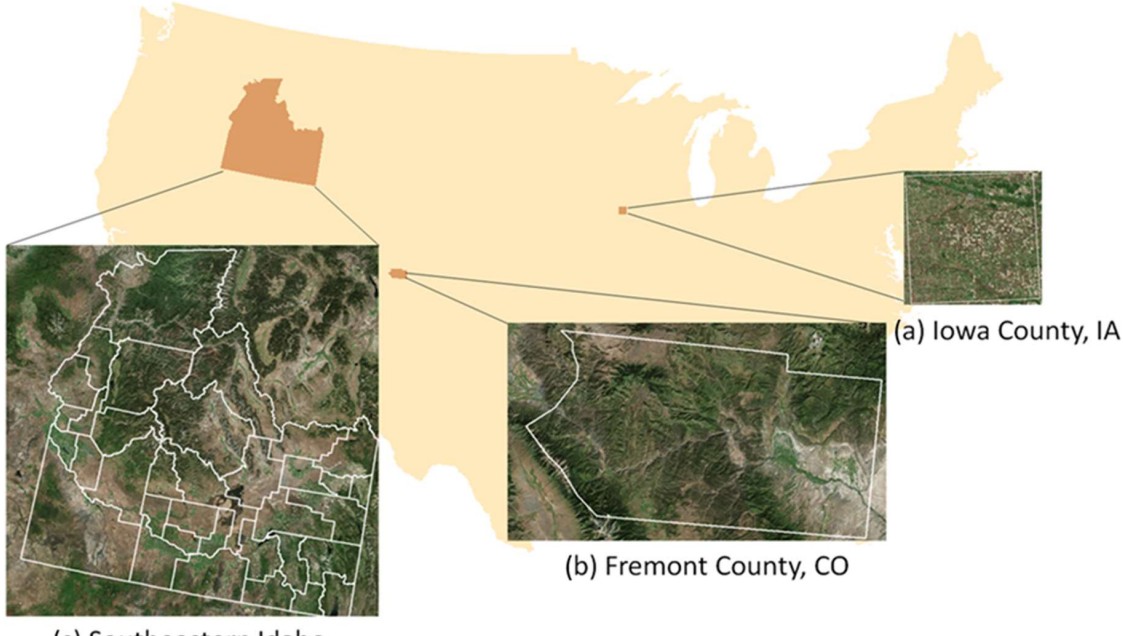

**Figure 2.** Selected area to represent three different landscapes: (**a**) flat land (Iowa County, IA); (**b**) mountain region (Fremont County, CO); and (**c**) mixed area (southeastern Idaho).

The annual flood dataset in 2013 was chosen for Fremont County and Iowa county for two reasons: (1) both counties experienced a large flood event in 2013 (Iowa county involved in 2013 Midwest flood from April to June, and Fremont County experienced 2013 Colorado flood in September respectively) [47], and using data from the same year will be able to keep the consistency of input data; (2) yearly flood information was used to ensure the dataset has both flood and a large amount of accumulated false classification within the whole year which benefits the result comparison later.

The flood frequency layer was used for southeastern Idaho since the annual frequency layer contains false classification for seasonal and yearly variation. It may not be easy to see false classifications from terrain shadow from daily flood maps or flood event maps, but the noise from terrain shadow could be seen clearly when looking at long-term data, for example the flood frequency layer. At the same time, flood frequency was suitable for this area since there was no reported flood event at this region between 2011 to 2015. In addition, the combination of different types of landforms (including flatland, mountain, rivers, lakes) in southeastern Idaho provides a result comparison and evaluation for a vast landscape.

Two types of flood data were examined for different geographic locations: the annual flood product was processed for county-level comparison; and flood frequency layer was generated to

evaluate flood information at the combined area. In general, annual flood layers which were supplied directly by the DFO represent if a location experienced a flood during a certain year. The data was generated by compositing daily flood data from 1 January to 31 December each year. For example, if a pixel experienced flood in a year, the pixel will be identified as "flood" no matter how many/when floods happened in that year. We then created a flood frequency layer from annual flood layers. Six annual flood layers (2011 to 2016) were combined into one layer using the "Union" function in Geospatial Data Abstraction Library (GDAL). "USA Rivers and Streams layer", which could be retrieved from Esri (http://services.arcgis.com/), was applied to mask out permanent water body in the dataset. However, in these research, we used dataset before removing the permanent water body since these rivers and lakes serve as reference in the comparisons and validation. Each location will have a value (0–6) to represent how many years it has experienced a flood. More technical details could be found in previous research [48]. Both annual flood layers and flood frequency data can be accessed directly or through web-services at: https://dss.csiss.gmu.edu/RFCLASS/.

Remote sensing generated a series of product describe the topology of the Earth which is called a digital elevation model (DEM) [49]. One of the most popular DEM datasets is called Shuttle Radar Topography Mission Digital Elevation database (SRTM). SRTM DEM data was generated by the Jet Propulsion Laboratory at California Institute of Technology (JPL) [49]. JPL released its global DEM product at 90 meters in 2003 [50]. The 90m SRTM DEM for the United Stated was downloaded to process the slope in this paper.

The 30 meter DEM data became free access to public as well after later 2015 [51]. However, the dataset is too large due to the fine spatial resolution. The large size of the dataset created a technical difficulty when generating slope, especially for the whole United States. Traditional ways of data processing were not able to serve the large volume of data. New cloud-based processing technique was utilized for generating slope data from 30-meter SRTM DEM. Google Earth Engine (GEE) is one of the most significant developments in the geoprocessing field in terms of processing speed [52]. Cloud-based technologies and geospatial web-services could bring the waiting time for geoprocessing from hours or days down to minutes or even seconds [53–58]. For example, generating slope data from 30 meter DEM at GEE platform took a few seconds while it takes serval hours to process coarser data (90 meter DEM) with the same steps on the workstation. GEE is extremely powerful, but it does not allow downloading of a large dataset although the data can be calculated and stored on the server. For this reason, a part of the processed slope data (southeastern Idaho) was downloaded to test if a change in spatial resolution of DEM filter will impact the result of proposed algorithm.

*2.2. Methodology*

A few preprocessing operations using Esri ArcMap were conducted to ensure the consistency between 30m and 90m slope data. Raster slope data was first generated from SRTM DEM data. After that, the slope was grouped into a few categories to reduce data size: less than 5°, 5–10°, 10–15°, 15–20°, larger than 20°. Classified raster was then converted to vector format. Finally, slope data was projected to the USA Contiguous Albers Equal Area Conic US Geological Survey (USGS) version to reduce distortion of the area in the data. Repeat the above steps for all 1 × 1° DEM tiles and merge the result into one layer.

A systematic framework of our method is present in Figure 3. The proposed algorithm considers both the impact of the slope and aspect on the terrain shadow to remove the misclassified flood area. Since the study area is located in the northern hemisphere, most of the terrain shadow appears in the northern direction. The aspect of north range (<90° and >270°) was first applied to extract the northern direction and then three different slope (>5°, >10°, and >15°) were applied to the NASA/DFO flood product to extract three refined flood areas as: flood area with slope <5°, flood area with the slope <10°, and flood area with the slope <15°. The final result was compared with the original NASA/DFO flood product and validated with the flood event data.

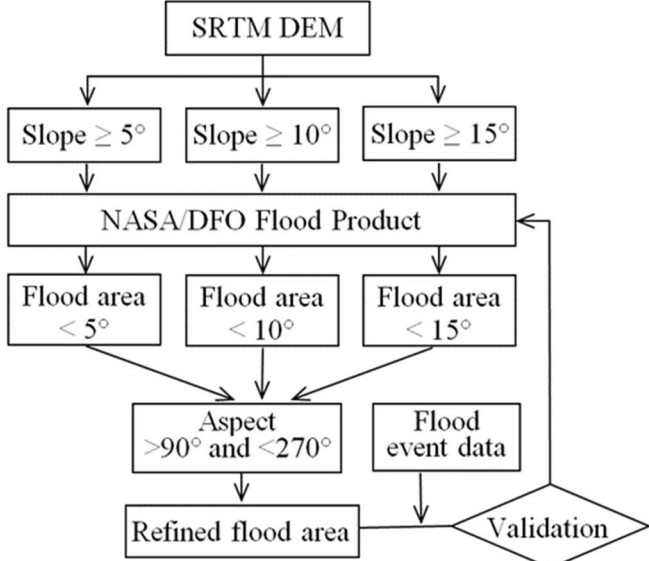

**Figure 3.** Flowchart of the proposed method.

Fremont County was selected to represent the mountain region (Figure 2), and it experienced the Colorado flood in September 2013. As this county is surrounded by a few mountains and peaks, certainly these valleys become a potentially higher risk for flood misclassification (Figure 4). Figure 3 shows the effects of the algorithm used to filter the misclassified area at 5° and 10°. In order to test this algorithm in a flat area, we selected one county at Midwest Iowa County, Iowa, during the flooding period between May and June 2013. We also conducted the same step for southeastern Idaho to represent combined mountainous and flatland areas.

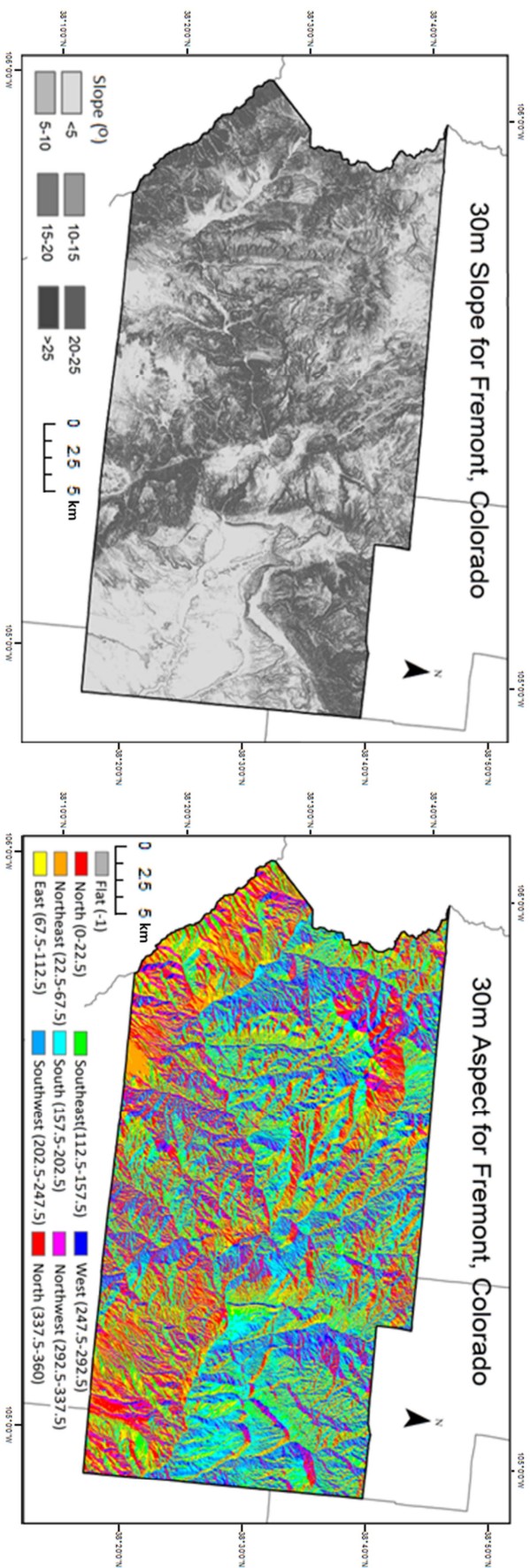

**Figure 4.** Slope and aspect image for Fremont County, Colorado.

## 3. Results

In Figure 5, some mountain shadow was illustrated when removing the flood area with a 10°
slope region and significant shadow has been removed with a 5° slope filter. Using a 10° slope region,
total flood area was decreased from 487 square kilometers to 285 square kilometers (more than 40%
reduction) while the 5° slope filter decreases the flood area from 285 square kilometers to 148 square
kilometers (more than 70% reduction). Most of the removed area was found in the ridge area and
the flooded valleys were not affected by this filter. The figure corresponds to the visual result which
indicates a significant removal of false classification in a water surface caused by mountain shadow.

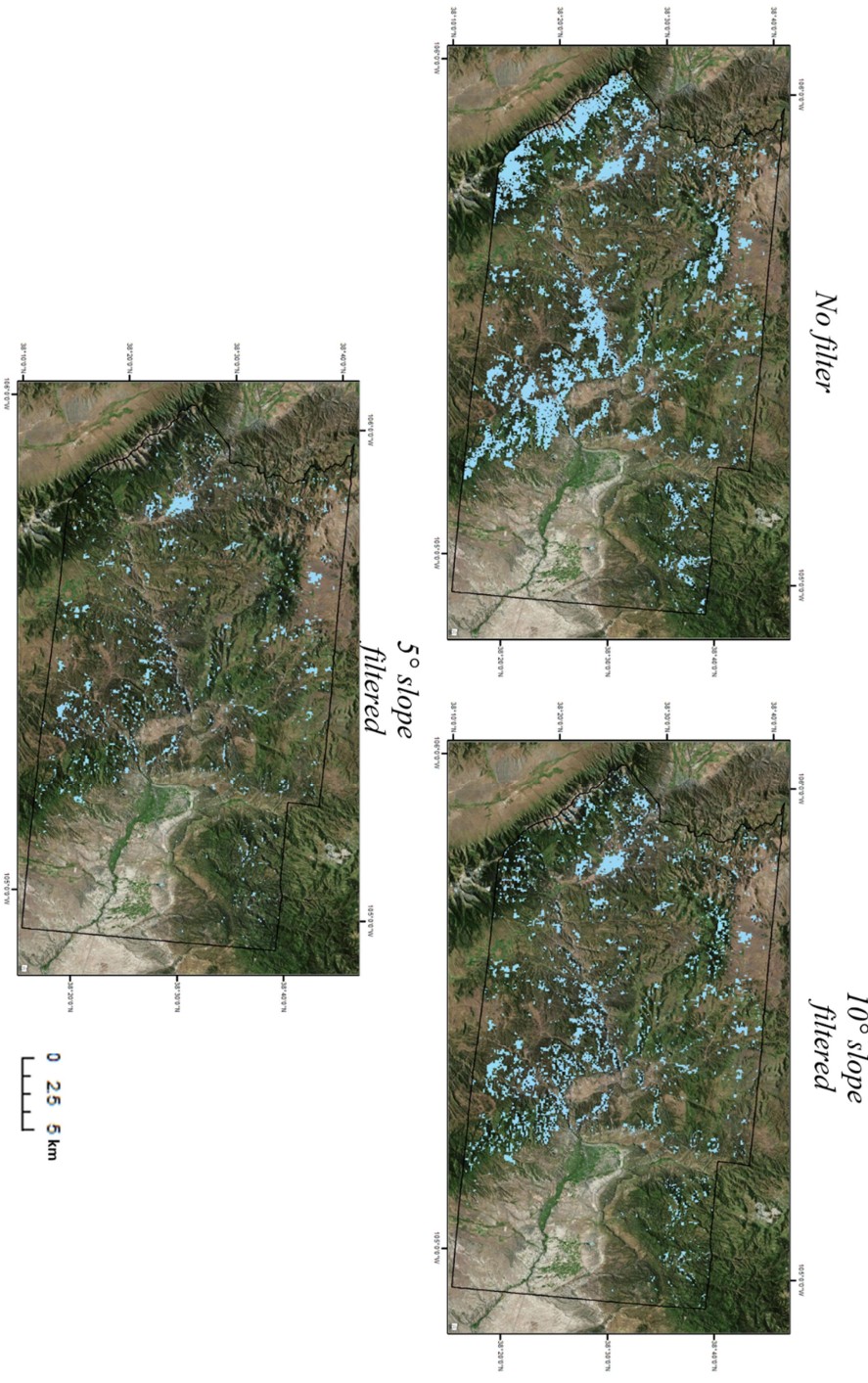

**Figure 5.** Flood area and filtered flood area using 5° and 10° slope (Fremont County, Colorado).

Figure 6 shows the results after applying 10° and 5° slope filters. It is easy to see that most of the flooded area is along the Iowa River, which has not been filtered out in the results map. The result indicates that the slope filter does not impact the flood area in the flat area.

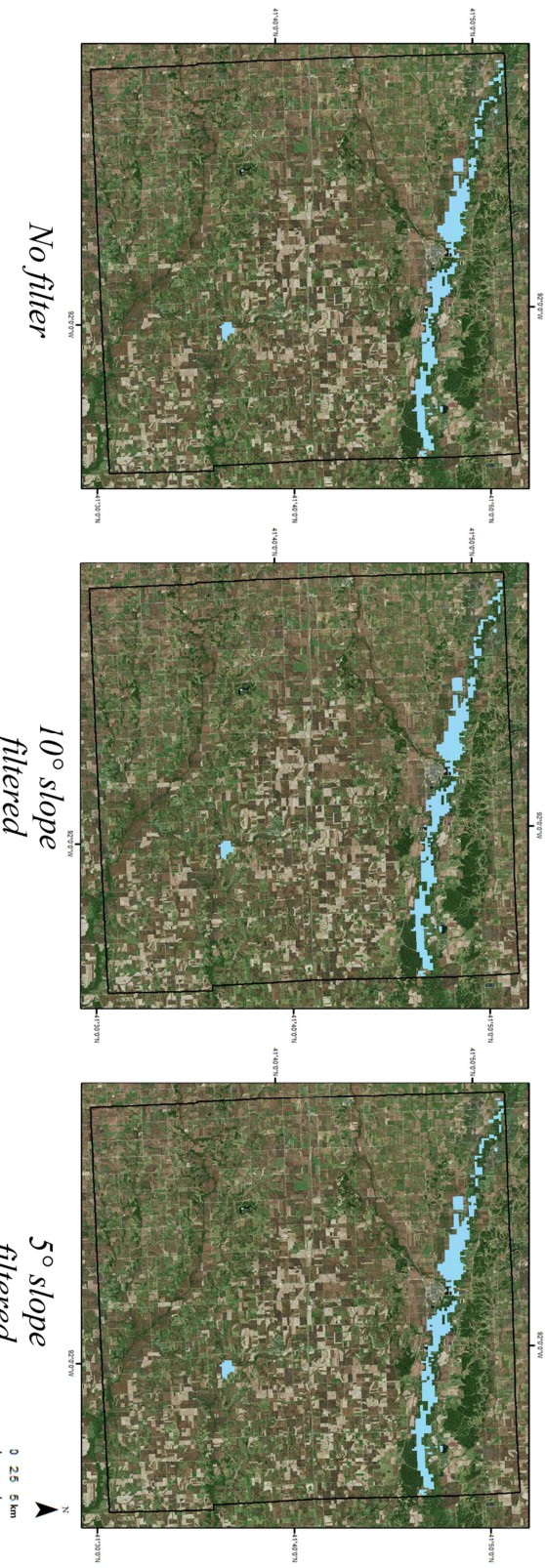

**Figure 6.** Flooded area and filtered results in Iowa County, Iowa.

The mixed mountainous and flatland region, southeastern Idaho, was selected to test our algorithm since it has had no significant flood since 2011 which indicates most of the flooded area in this region were mostly a false classification. In Figure 7, a considerable misclassification was removed (50%) using the 10° slope filter while another 17% flood area was removed when applying the 5° slope filter, mainly distributing in mountainous area.

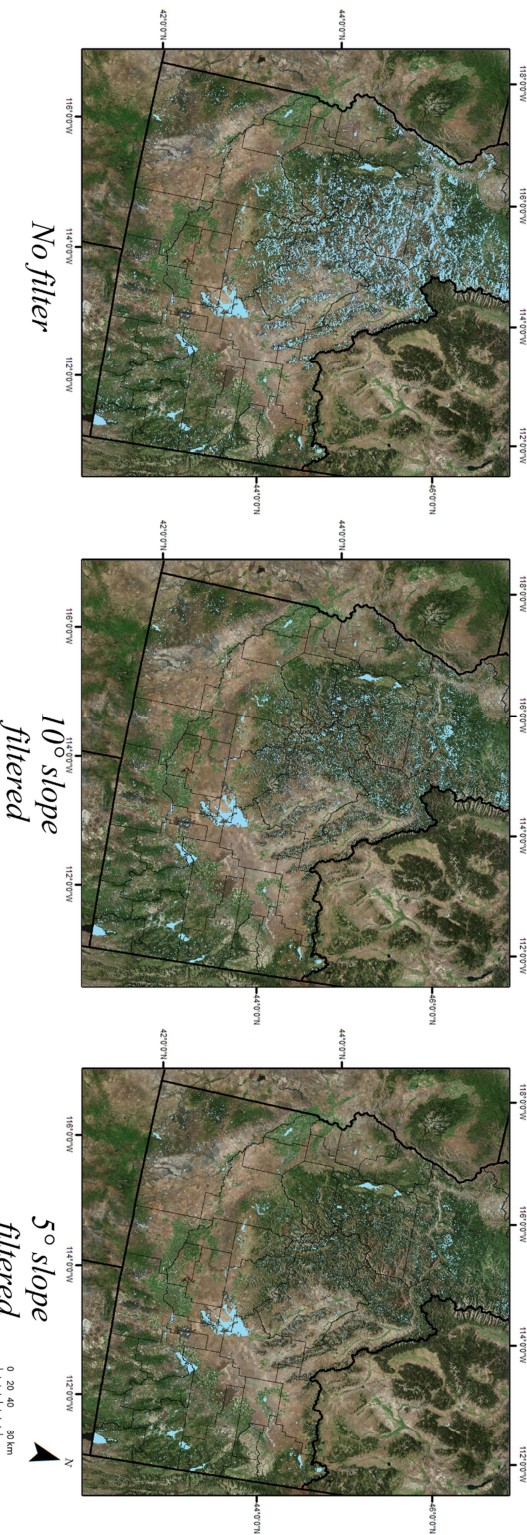

**Figure 7.** Water surface and filtered false classification by 30 m slope in southeastern Idaho.

In order to test the impact of resolution, we generated a coarse slope layer (CSL) from 90 m STRM DEM data and compared the filtered result with the fine slope layer (FSL) in southeastern Idaho (Figure 8). We can notice that the filtered results from CSL and FSL are different from each other, reducing more than 55% the misclassification of flooded area by the 5° FSL. Compared with the original data, more than 80% of flood area was discarded and the significant reduction was focused within the mountain area which was consistent with the "actual" flood area.

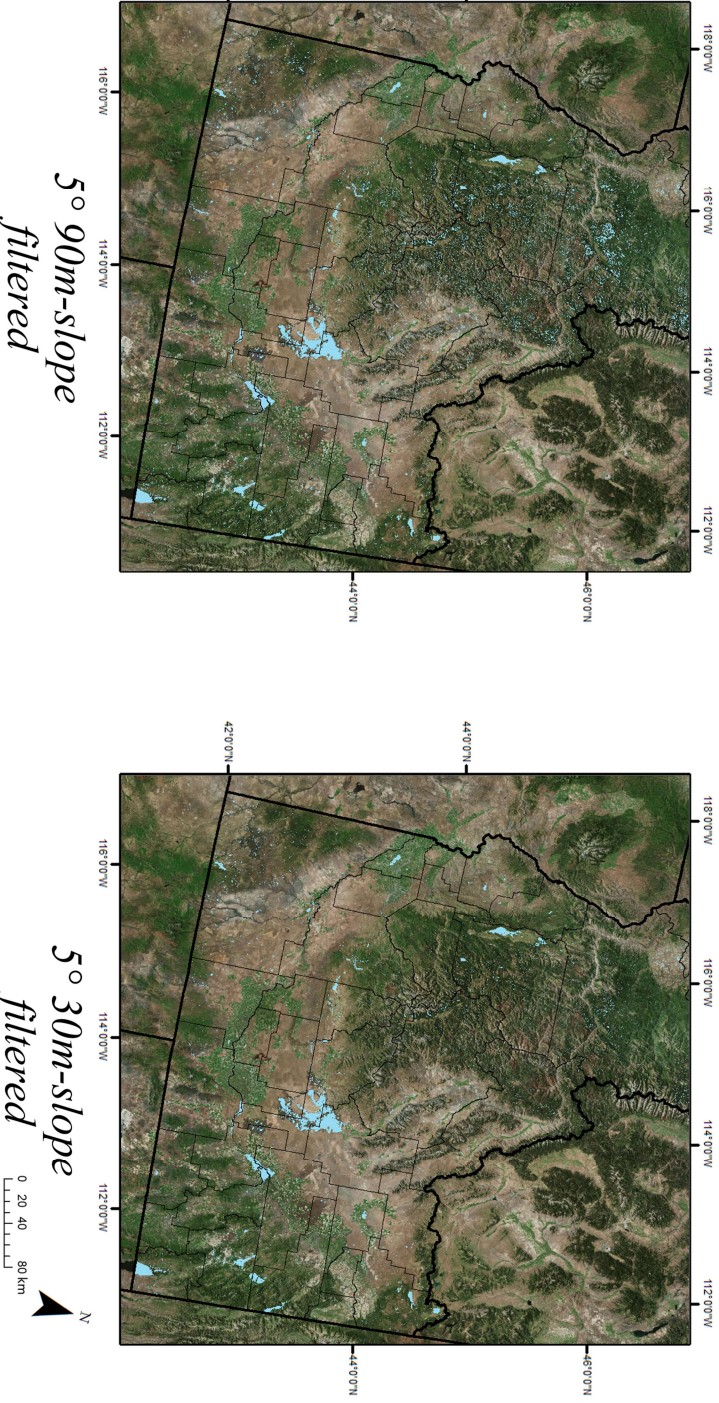

**Figure 8.** Water surface and filtered results from 5° slope data at 90 m and 30 m digital elevation model (DEM).

In the northern hemisphere, it is generally assumed that more terrain shadow will be generated in the northern direction. More than 40% of the flooded area was found in the northern direction (aspect >90° and <270°) while other three directions have a relatively even distribution (20% of total area in each direction) (Table 1). After applying the slope filtering, we could find most of the removed area (44%, 3614km$^2$) is in the northern direction, and the distribution of flooded area in four directions is rational (31% in the north, 22% in the east, 27% in the south, and 20% in the west) after the filtering (Table 1).

**Table 1.** The total flooded area, removed area, and filtered results in four directions.

| (Unit: km$^2$) | North | East | South | West | Total |
|---|---|---|---|---|---|
| Total flood area | 4537.99 | 2405.00 | 2153.97 | 2113.58 | 11210.54 |
| 15° slope filter | | | | | |
| Removed area | 1708.13 | 725.30 | 533.24 | 732.95 | 3699.62 |
| Filtered results | 2829.87 | 1679.70 | 1620.73 | 1380.62 | 7510.92 |
| 10° slope filter | | | | | |
| Removed area | 2652.36 | 1212.10 | 882.95 | 1107.36 | 5854.77 |
| Filtered results | 1885.64 | 1192.90 | 1271.02 | 1006.21 | 5355.77 |
| 5° slope filter | | | | | |
| Removed area | 3614.35 | 1765.68 | 1360.94 | 1531.80 | 8272.77 |
| Filtered results | 923.64 | 639.31 | 793.03 | 581.79 | 2937.77 |

A 3D animation was created using ArcScence to show the result after removing false classification due to mountain shadow. Figure 9 illustrates the result after removing false classification for a part of southeastern Idaho by taking a few screenshots of the 3D animation at different viewing angles. Grey areas indicate the terrain, pink regions are the false classification identified by our method, and blue is the water bodies identified by our method. In order to reduce the animation size and provide the best user experience, only part of southeastern Idaho was used to create the animation. The animation not only shows the difference between different aspects but also supports the fact that most false classification due to terrain shadow could be removed by the proposed method. Although not all false classification were identified, we can see a significant amount of mountain shadow was successfully separated from true water bodies.

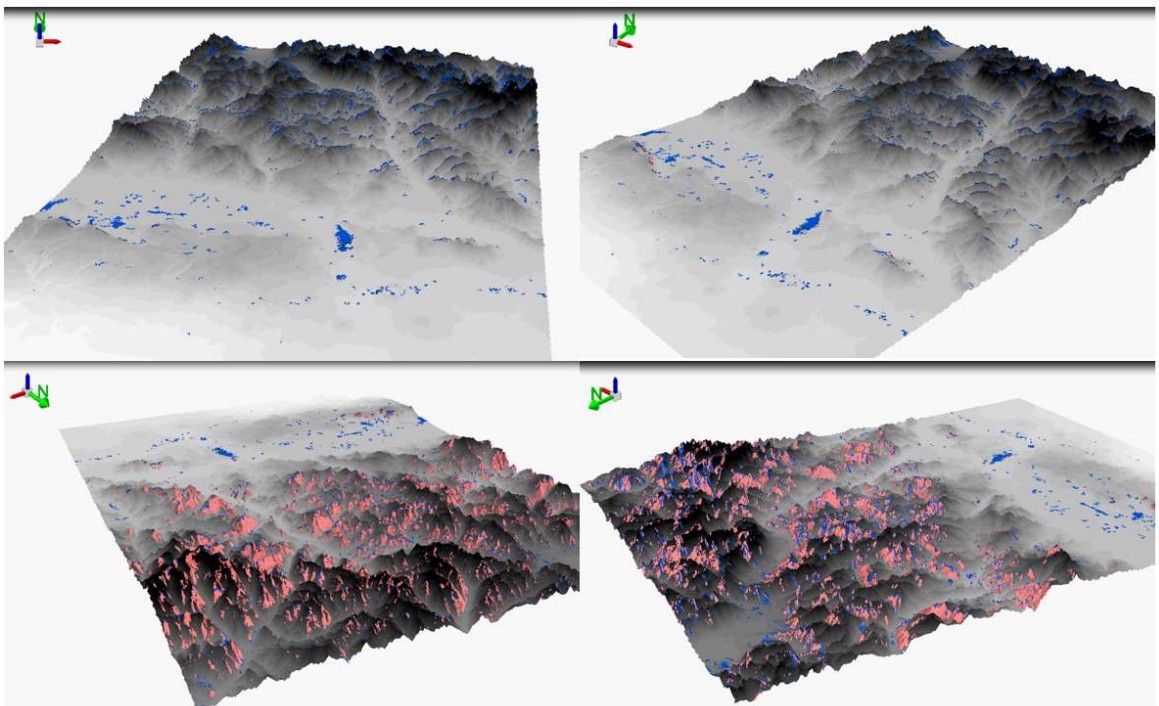

**Figure 9.** 3D animation to show the distribution of false classification identified by our algorithm. (part of southeastern Idaho; pink indicates area being removed by the algorithm, blue indicates remaining flood area in the dataset).

We selected Valley County in Idaho to conduct a validation and accuracy test since it contains few water bodies which could be used as a reference, and no flood event during the study period helped to validate false classification. The result from three filters was compared with original flood data. Since there were no flood events in the study area, it is easy to distinguish if a pixel is removed correctly. First of all, we measured the area of water surface and false classification, then we compared results from each filter to calculate the amount of false water being correctly removed, and the amount of water being incorrectly removed.

Within a total 2650 square kilometer flood area in the validation area, there are 210 square kilometers of water surface and 2440 square kilometer of false water. With different filter methods, about 74% of the total area were removed by the 15° filter, 82% of entire area were removed by the 10° filter, and 90% of the entire area was removed by the 5° filter. For false classified water area, about 80%, 88% and 96% of the area were correctly removed by the 15°, 10°, and 5° filters, respectively. In addition, the filter also contributed to some error in removing the true water surface which accounts for 4%, 8%, and 17% for the 15°, 10°, and 5° filters, respectively (Table 2). Sensitivity, specificity, false positives, and false negatives are shown in Table 3.

**Table 2.** Gold standard test for different filters (Valley County, Idaho).

| Slope Filter | Test | Gold Standard | |
|---|---|---|---|
| | | **Not-Water** | **Water** |
| 15° | not-water | 1938.172 km$^2$ (TP) | 6.542 km$^2$ (FP) |
| | water | 501.921 km$^2$ (FN) | 203 km$^2$ (TN) |
| 10° | not-water | 2145.758 km$^2$ (TP) | 16.786 km$^2$ (FP) |
| | water | 294.336 km$^2$ (FN) | 193.214 km$^2$ (TN) |
| 5° | not-water | 2335.774 km$^2$ (TP) | 35.487 km$^2$ (FP) |
| | water | 104.320 km$^2$ (FN) | 174.512 km$^2$ (TN) |

**Table 3.** Sensitivity, specificity, predictive values, and overall accuracies (Valley County, Idaho).

| Slope Filter | True Positive Rate | False Negative Rate | False Positive Rate | True Negative Rate | Positive Predictive | Negative Predictive | Overall Accuracy |
|---|---|---|---|---|---|---|---|
| 15 ° | 0.794 | 0.206 | 0.031 | 0.969 | 0.997 | 0.288 | 0.808 |
| 10 ° | 0.879 | 0.121 | 0.080 | 0.920 | 0.992 | 0.396 | 0.883 |
| 5 ° | 0.957 | 0.042 | 0.169 | 0.831 | 0.985 | 0.626 | 0.947 |

## 4. Discussion

Shadow and water share a similar spectral pattern, so it is hard to distinguish them in a remote-sensing approach. Previous research has addressed some misclassifications to improve the product's quality by removing cloud shadow and part of the terrain shadow. However, these approaches cannot identify permanent shadow from the terrain. By introducing DEM to describe topology, this paper introduced a new way to improve the DFO flood product.

Comparison experiments in this study suggested that the proposed algorithm could correctly identify false classifications from terrain shadow in mountain regions. In addition, the algorithm has no impact on the selected flatland study area. Following comparison experiments in small study areas (Fremont County, Colorado and Iowa City, Iowa), the algorithm was adopted in the larger study area (southeastern Idaho) and result supported the conclusions from the small study area. Furthermore, finer spatial-resolution DEM was used in this study to test if there is any difference in result. Results from both statistical and visual comparisons indicated that slope filter from the 30 meter DEM leads to better results than the 90 meter DEM.

In addition to comparing area removal, accuracies were assessed for the selected study area (Valley County, Idaho). The result indicated an excellent identification of false classification by using different filters. However, these filters performed differently in detecting water surface. Furthermore, the overall accuracy increased significantly from using 15° to 5° slopes.

There were some limitations to the work that should be addressed in the future. First, the experiment assumed that all flood area in the mountain is misclassified, which could introduce some error. Currently, there is no good flood product that can be used in large spatial extent and fine temporal resolution research. For this reason, it is hard to find a third party resource to validate the filtered result generated in this study. We tried to reduce this error by using an area that has no flood record during the experiment period. We have also validated the accuracies in the result using subset images. However, small flash floods could happen during the process by which images were taken but they may not be recorded since there was no damage to properties. Second, the slope method is limited to remove shadow caused by terrain, while it is not able to remove the shadow from other sources such as cloud. Some noises were still existing and needed to be further processed. Third, although we can see significant removal of flood area in the mountain, the shadow was not removed completely. The experiments concluded that more flood areas were removed when introducing a higher-resolution slope filter. However, the cutoff value for slope was arbitrarily chosen to test the feasibility in the research, and so this does not represent the best choice. It is noticeable from the accuracy assessment that the false positive rate increased about 13% while moving the filter from 15° to 5°. This indicates that more true water surface could be incorrectly removed by the proposed methods. The research did not use a filter higher than 30m due to the limited computing resource; also, we may not obtain better results from increasing resolution in the slope. More research is needed to find the best threshold for filtering out mountain shadow.

## 5. Conclusions

RF-CLASS was developed by CSISS to provide both flood information and crop loss estimation from flooding for the United States. This information is currently serving as one of the primary sources

for the US Department of Agriculture (USDA) Risk Management Agency to evaluate crop insurance. As a result, the accuracy of the flood information is critical.

This paper aims to provide a new way of improving flood data by including slope to describe topology. The slope was proposed to help reduce flood false classification for two reasons. First of all, slope describes the potential risk of flood. Since water always flows from high to low, the risk of flood declines when increasing slope. Secondly, as the slope increases, larger shadow will be produced. Based on these two assumptions, this paper tries to separate shadow from flood data in order to improve the accuracy of the flood result.

CSL was generated from a 90-meter resolution SRTM DEM layer. FSL was produced and downloaded from Google Earth Engine. After determining a few experimental slope filters, slope mask layers were created. Both CSL and FSL were used to extract mountain shadow from the flood layers. The experimental result suggested that the area of water surface was reduced by applying slope filters.

Few comparisons were developed in order to test the feasibility of removing false classification from mountain shadow using the proposed method. Both small (county-level) and large (state-level) scale comparison were developed. Small-scale comparisons could be grouped into two categories: flatland region and mountain region. Figure 5 & 6 pointed out that there is no change in the flood area in flatland while the significant decline of flood area occurred in the mountain region. The reduction of flood area was not randomly distributed when visually inspecting the results. A noticeable amount of reduction happened in the mountains while the change on the flatland was limited. Also, adding aspect as another input provided a more accurate way to identify false classification of the mountain shadow.

A high percentage in both sensitivity and specificity is needed to check if an algorithm is good enough to filter out false classifications. As we move the filter from $15°$ to $5°$, a trend of decreasing sensitivity and increasing specificity could be identified. Among these three filters, the test suggests $5°$ performed better than the other two filters by providing the highest overall accuracy. Thus, the paper concluded that it is feasible to use the proposed algorithm to highlight mountain shadows with low flood risk, and thus remove pixels misclassified as water. Moreover, the improvement of spatial resolution of the slope filter contributed to removing more noise from the flood layer in detail.

**Author Contributions:** Conceptualization, L.L.; Data curation, L.L.; Formal analysis, L.L.; Funding acquisition, L.D.; Investigation, L.L.; Methodology, L.L., L.D., E.Y., M.S.R., R.S. and L.K.; Project administration, L.D.; Resources, L.L.; Software, L.L. and C.Z.; Supervision, L.D.; Validation, L.L.; Visualization, L.L.; Writing–original draft, L.L.; Writing–review and editing, L.L. and J.T.

**Funding:** This research was funded by grants from NASA Applied Science Program (Grant # NNX14AP91G, PI: Prof. Liping Di) and NSF INFEWS program (Grant# CNS-1739705, PI: Prof. Liping Di).

**Acknowledgments:** The authors would like to take this opportunity to thank members of the editorial board and anonymous reviewers for their helpful comments and suggestions.

**Conflicts of Interest:** The authors declare no conflict of interest.

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
