# Peer review of "Improvement and Validation of NASA/MODIS NRT Global Flood Mapping"

_remotesensing, doi:10.3390/rs11020205_

Round 1

Reviewer 1 Report

It can be accepted in present format. Just make sure that it follows all the policies of the journal.

Author Response

Dear Reviewer,

Thank you for your comments.

Please see the reply from attached document.

Best,

Li Lin

Reviewer 2 Report

Generally, the reply to reviewer should contain all the comments of the reviewer and the answer to all the comments of the reviewers. This is not the case. In the reply of the reviewer it is expected that the authors mention the changes in the text in order to facilitate the reviewer to find them.

The authors did not improve the paper and the analysis in general. Only a few comments are considered during this first round of revision that in my opinion, do not change the quality of the paper.

Referring to comment 1, no mention is given about the origin of the flood data and the flood frequency. More information is requested about the analysis on floods. Lines 146-148 are not sufficient to explain how to get the flood information. Only a flood event in 2013 is considered. How did the authors use the information about the frequency? This part is not clear at all. About the reference to the paper of Li (2017), I have to underline that a paper should be completed without reading another paper to understand the analysis. Therefore, the authors should provide the main steps of this phase that is completely not understandable.

Comment 2 had many questions and the authors replied only to some of them. I think the authors should specify the replies and describe the steps of the analysis in the manuscript. Again, it is not clear why two DEM are used and how.

In my opinion, the figures with flooded areas are not visible in a 100% of zoom (same size of the A4 paper). Only with a zoom of 300-400% it is possible to see some differences in the comparisons.

I do not understand the concept according to which the validation is carried out when a flood event is not occurring. If no flood event is observed in the study areas, why the information of flood is given as an input? Probably, the input is not reliable and should be reasonably justified. Indeed, authors said that Southeastern Idaho is selected to test the algorithm in flat and mountainous region, but no flood event is observed during the study period. So, what is the sense to select a region with no floods for testing an algorithm for floods?

Authors should describe also another case: when a flooded area is covered by a slope shadow. In this case, the filter could remove correct flooded areas. This is the case of the false negative that is present also in Table 2. A discussion about this point is required.

The English of the paper should be accurately revised. Many typos are present and some sentences should be reformulated.

SPECIFIC COMMENTS:

Line 28-29: Flood is one of the most widespread and frequent natural disaster. Please reformulate the sentence.

Di et al., 2018 is repeated in the introduction. To identify the citations, please use Di et al., 2018a and Di et al., 2018b.

Line 53: MODIS has also the limitation of the spatial resolution (250 m) that should be cited in the paper.

Lines 71-75: After the application of the algorithm, the authors should provide a discussion about the effectiveness of the proposed approach with respect to the original one. Moreover, the example of the inundated region in Southeastern Idaho at 2011 is not clear in the results session. As results, more flooded areas are provided. Please rearrange the sentence and the results.

Lines 78-87: If the purpose of the study is to improve and validate the MODIS-derived global flood mapping products, the authors should provide some more details about the advantages and the limitations (for example for which river width can be used) of the NRT global mapping.

Lines: 89-110: authors described different method to remove the cloud shadow, but is one of them applied in the analysis? If yes, which one? If no, how did you manage the errors induced by clouds?

Lines 163-178: why did the authors use two DEM? Actually, MODIS pixel is 250 m, therefore, 90 m can be sufficient to have a more reliable map of flooded areas. Please specify the reasons in the text.

Figure 5: because of the size of the figure is not possible to see the river and the observed flooded areas for comparison. It is quite obvious the reduction of the flooded areas with the slope (5° or 10°). The interesting part in my opinion is to see a river and if the flooded area around is correctly predicted.

Lines 258: in the introduction, the authors said that the validation is carried out in the Southeastern Idaho, but finally, the validation is conducted in Valley County. Please, be consistent.

In Table 2 the observations and the results of the algorithm are not clear. Moreover, the analyzed study area should be specified in the caption.

Author Response

(The authors gave the same response as above.)

Reviewer 3 Report

The author correctly addressed comments of first revision. However, some issues should be  fixed and clarified and revision of format is needed

> Affiliation needs to be corrected according to RS  journal format

> References need to be formatted to  RS journal format:   [1], [2] 

> Be consistent with units: meters/miles/kilometres are used along manuscript. I suggest using the metric system

> Please add a more precise reference to the NRT data used so it is possible to replicate the experiments.    (e.g. https://floodmap.modaps.eosdis.nasa.gov/getTile.php?location=110W050N&day=273&year=2013&product=14) 

> Take into account the possibility to create a classified  shadow model (using a proper Sun Azimuth )  to filter the data 

> Take into account the comparison with not flooded images as a possibility to filter false positive flood area related to mountain shadow.   

Author Response

(The authors gave the same response as above.)

Round 2

Reviewer 2 Report

I thank the authors because I really appreciated the reply to each comment I provided on the previous round of review.

I solved most of my doubts even if a couple of questions still remain.

I am sorry because I repeat always the same question, but I re-read the paragraph 3.2 at least ten times and I really does not understand how the flood events have been extracted and the flood frequency layer has been generated. I also read the paper of Li et al (2017) the situation does not improve. I think the problem can be solved providing more information. Indeed, authors say “The data was generated by compositing daily flood data from January 1st and December 31st each year.” (Lines 166-167). What did the authors mean with “compositing”? Is it a sum? If yes, each pixel can assume a value between 0 and 365 (or 366) and a sort of flood frequency can be calculated. Successively, the six annual flood layers were combined into one layer (how? Again with a sum?) and for each location it is possible to have a value between 0 and 6. Exactly what does it mean? A pixel that includes a river is always wet and flooded, hence, the flood frequency should be 6 (it is permanently wet), but similarly, if a close pixel is inundated once per year (for example during a flood season) the same value is generated. Is this interpretation correct?

About the DEM, was the processing from line 191 to line 196 carried out for both the DEM? If yes, please specify. Moreover, please add also a sentence explain the reason to analyze 30 m and 90 m DEM.

About the figures with flooded areas, I they should be bigger to ensure the view without zoom (if you read the paper not in digital form, you should be able to see all the content). But I cannot insist on this point, if the editor agrees with the choice of the authors.

Line 55: replace "compared" to "compare".

Author Response

Thank you for the constructive comments and suggestions. Your comments and suggestions contribute a lot to the revised version of the manuscript. We followed the recommendations and responded to each comment below.

General comments

1.       I re-read the paragraph 3.2 at least ten times and I really does not understand how the flood events have been extracted and the flood frequency layer has been generated. I also read the paper of Li et al (2017) the situation does not improve. I think the problem can be solved providing more information. Indeed, authors say “The data was generated by compositing daily flood data from January 1st and December 31st each year.” (Lines 166-167). What did the authors mean with “compositing”? Is it a sum? If yes, each pixel can assume a value between 0 and 365 (or 366) and a sort of flood frequency can be calculated.

ANSWER. In fact, the annual layer was generated and provided by DFO (added in line 167-168). The annual flood layer and some other products are only available on our application (RFCLASS). For this reason, in this paper, we only discussed the general idea of how the annual flood layer was generated (line 168 - 171). In general, in remote sensing compositing refers to the process of combining spatially overlapping images into a single image based on an aggregation function. Here the raster composition is counting area aggregation and not considering any value summation (line 170).

2.       Successively, the six annual flood layers were combined into one layer (how? Again with a sum?) and for each location it is possible to have a value between 0 and 6. Exactly what does it mean? A pixel that includes a river is always wet and flooded, hence, the flood frequency should be 6 (it is permanently wet), but similarly, if a close pixel is inundated once per year (for example during a flood season) the same value is generated. Is this interpretation correct?

ANSWER. This process was conducted by us. We used GDAL to union the annual flood layer together (line 172). It is very like the Union tool in ArcGIS. The difference is this is for the rater. The idea is just counting frequency for each pixel and sum the result. In this case, we have total 6 years the max/min would be 6 and 0 (line171-172, line 175-177). Regarding the concern of permanent water bodies, we removed them using a reference water body layer in the final products. However, in these research, we used dataset before removing permanent water body since these rivers and lakes serve as reference in the comparisons and validation. (line 161-164)

3.       About the DEM, was the processing from line 191 to line 196 carried out for both the DEM? If yes, please specify. Moreover, please add also a sentence explain the reason to analyze 30 m and 90 m DEM.

ANSWER. Yes, this has been added to line 192-193. The reason to explain why two DEM datasets were used could be found at line 196-197

4.       About the figures with flooded areas, I they should be bigger to ensure the view without zoom (if you read the paper not in digital form, you should be able to see all the content). But I cannot insist on this point, if the editor agrees with the choice of the authors.

ANSWER. We agree with the reviewer’s comment. In addition to word and table comparisons, the figure itself should be clearly presented. So, we changed the direction of the figure and scaled to the whole page (page 8&9). This should provide better visualization while it may not be the best solution. Also, we can upload the original images for readers to download if this is an acceptable solution.

Specific comments

1.       Line 55: replace "compared" to "compare".

ANSWER. I changed “compare” to “compared”

Reviewer 3 Report

The authors correctly addressed most of the comments of first revisions. However, the papers steel need major  revision because the  manuscript still not match the  Remonte  Sensing paper structure : 

Please provide the division of  chapter according to RS Journal (https://www.mdpi.com/journal/remotesensing/instructions)  with the following necessary chapters: Introduction,  Materials and Methods,   Results, Discussion  and Conclusions

I suggest to reorganise the manuscript and in particular: 

1. Move Paragraph 3.2 o section  4. Section 4 should be considered the section " Materials and Methods"

2.  Separate discussion from results 

3.  Join the chapter 7. future work to conclusions 

*****************************

Other minor issue 

Some considerations that can be done:

MODIS has a spatial resolution of  250 / 500 m and most of the small floods that occur in the mountains cannot be detected.  

For single a flood event, the effect of the topographic shadow changes a lot depending from the sun inclination (latitude and day of the year). 

Author Response

Thank you for the constructive comments and suggestions. Your comments and suggestions contribute a lot to the revised version of the manuscript. We followed the recommendations and responded to each comment below.

General comments

1.       I suggest to reorganise the manuscript and in particular: 1. Move Paragraph 3.2 o section  4. Section 4 should be considered the section " Materials and Methods"2.  Separate discussion from results 3.  Join the chapter 7. future work to conclusions

ANSWER.  Paragraph 3.2 has been moved to section 2-“materials and methods”. Removed background section and split into section 1-“introduction” and 2-“materials and methods”. Section discussion has been separated from result, and some more discussions have been added to section-discussion. Moved future work to discussion according to the template.

2.       MODIS has a spatial resolution of  250 / 500 m and most of the small floods that occur in the mountains cannot be detected.  

ANSWER. Discussion on MODIS’s spatial resolution has been added to line 55-57; 83-84; and discussion on small flash flood in mountain located between line 327

3.       For single a flood event, the effect of the topographic shadow changes a lot depending from the sun inclination (latitude and day of the year).

ANSWER. Yes. These parameters (time, date, etc.) are needed to be considered in other methods (line 88, 109), and these have been done in the past. However, significant mountain shadow still exist in the product (e.g., the original image in figure 7), this is why the proposed algorithm is important. It may not be clear to see false classifications in daily flood map, but when looking at long period data, for example, annual flood data or annual flood frequency, the noise from terrain shadow build up. For this reason, we used the flood frequency layer. (discussion added to line 157 -161)